# Vibration Measurements on a Six-Axis Collaborative Robotic Arm—Part I

**DOI:** 10.3390/s23031629

**Published:** 2023-02-02

**Authors:** Vit Cernohlavek, Frantisek Klimenda, Pavel Houska, Marcin Suszyński

**Affiliations:** 1Faculty of Mechanical Engineering, University of Jan Evangelista Purkyne in Ustí nad Labem, Pasteurova 1, 40096 Ustí nad Labem, Czech Republic; 2Institute of Mechanical Technology, Poznan University of Technology, 60965 Poznan, Poland

**Keywords:** six-axis robotic arm, vibration and noise, dominant frequency

## Abstract

This article deals with the design of a methodology for vibration and noise measurement on a six-axis collaborative robotic arm. A vibration and noise measurement methodology is proposed for six robot positions. In each position, measurements were performed under defined equal boundary conditions. The boundary conditions were related to the velocities of the joints and the load on the robotic arm. The second part of the article is an evaluation of the initial experimental results. So far, only the acceleration of the sixth joint of the robotic arm—Wrist 3—has been measured. The aim of the measurements was to verify if the methodology presented can be used for vibration measurements. From the evaluation of the experimental measurements, it was determined that the given methodology can be used for vibration measurements. It was also found that the acceleration is transmitted in the axes other than the axis of motion of the robotic arm. In future experiments, the vibration at the other joints of the robotic arm will be measured and the noise of the robotic arm will be measured to confirm whether the proposed methodology is indeed functional.

## 1. Introduction

Vibrations are a phenomenon that we encounter every day, as all bodies or particles of movement are a source of larger or smaller vibrations. These vibrations are so small that they are not measurable in any available way, and their existence can only be predicted on the basis of a mathematical–physical model, as well as the vibrations that we are able to perceive simply by touch without any devices. Vibrations can be profitable for humans such as when used in mobile phones, sieving tables in mills, electric and pneumatic hammers, compaction and vibration equipment for road construction and maintenance, or loudspeakers. After all, sound is created by vibrations. However, there are often situations in which vibrations are undesirable, such as automatic washing machines, which are a source of unpleasant noise. In engineering, where noise is not so carefully handled, vibrations may be harmful, as components of different machines and devices are subject to the influence of greater loads, and with these, their limits of fatigue decrease. In several cases of bridge collapses, vibrations and an associated phenomenon—resonance—are also to blame. Last but not least is the most dangerous manifestation, an earthquake. One soon realizes that vibrations need to be understood, and their principles found and proven [1,2].

Noise and vibrations are harmful, and the majority of the contemporary human population encounters them almost every day and anywhere. They can be described as a phenomenon of the last decade, and it has been shown that their amount in the environment is constantly increasing, which is related, among other things, to the increase in new technologies, tools, or devices in society. At present, few can say that they have never felt exposed to a high amount of noise stress or vibrations. The exposure of persons to excessive noise or vibrations has caused, inter alia, an increase in legal disputes between persons harassed by noise or vibrations on the one hand, and persons producing noise or vibration as a result of their activity on the other. In such disputes, there are rights to protect the health or the environment in play along with the right to entrepreneurship or the effort for economic progress and the development of society. It follows from the above that due to the competition of these protected interests, it is necessary to offer a legal order such as one to achieve balanced protection of all concerned and concurrent rights. For this reason, the legislation on noise protection and vibrations is presented as a significant part of the European Union’s legal order [3,4,5].

Vibration in technical literature [6] is considered to move a flexible body or environment whose individual points oscillate around their equilibrium position. The definition of vibration contained in the Public Health Protection Act in Section 30 (2) deals primarily with the harmful effect of vibration on humans, and vibration is defined as follows [6]: “Vibration means vibrations transmitted by solid bodies to the human body that can Being harmful to health and whose hygienic limit is set by the implementing legal regulation”. Vibration does not spread, unlike noise, through the air, but above all, it spreads through solid structures. Vibration sources can be divided into stationary and non-stationary, depending on the type of signal. Stationary signals can then be further divided into deterministic (periodic) and random ones. For example, with vibrations from industrial production, machines that produce these vibrations can be considered stationary deterministic signal sources. A stationary random signal may be the occasional passage of a train around a human dwelling, and this passage is repeated at certain time intervals. It is these two types that are important for human protection from the adverse effects of vibration. The third type is nonstationary random signals. Mostly in terms of protection against vibrations, we do not investigate random signals causing vibrations, which include earthquakes or vibrations caused by a neighbor hammering a picture into a wall. The increase in vibrations, as well as noise, is closely related to the increase in diameter production and increasingly demanding machines and equipment. However, vibrations are not as common as noise, and we usually encounter them in connection with excessive noise. Vibration also causes negative reactions in the human body, and we can say that they participate in the development of the same diseases as noise. In combination with noise, they still show their own negative effects [5,6,7,8].

A single degree of freedom is the most common in the case of an oscillating system. For vibration measurements, the most used are so-called accelerometers that convert acceleration to an analog electrical signal. These sensors are divided according to physical phenomena that take place in active action in determining the output quantities.

The vibration of machinery can be measured by different methods that offer more or less information about the condition of the machine. The simplest and most widespread vibrodiagnostic method is to measure the total vibration value. The effective value of vibration speed in the range 10–1000 Hz (2–1000 Hz) is measured according to CSN EN ISO 8041 [9].

Vibrations are a manifestation of mechanical oscillation. When measuring, we need to know whether we want to measure absolute or relative vibrations. Absolute vibrations describe the movement of the monitored body toward the Earth relative to us due to the base created (e.g., machine frame). Mechanical oscillation expresses a relationship [10,11,12,13,14]
(1)$$m\ddot{y}+b\dot{y}{+ky=F}_{b}$$
where *m*—weight of the seismic mass of the sensor, *y*—deflection, $\dot{y}$—velocity, $\ddot{y}$—acceleration, *b*—damping, *k*—stiffness coefficient, and *F_b_*—excitation force.

This article is devoted to a detailed description of the proposed vibration and noise measurement methodology for the UR10 six-axis collaborative robotic arm. The first part of the proposed measurement methodology is devoted to the description of the vibration measurement of each of the six joints of the robotic arm. For each joint of the robotic arm, a measurement position is described in which only one of the six joints of the robotic arm is always under stress. Thus, a total of six measurement positions are described. The second part of the proposed methodology is devoted to the description of the measurement of the total noise of the robotic arm in two positions. These are the *X*-axis and *Y*-axis noise measurement positions. In addition, the article of the proposed methodology can be applied. These are acceleration measurements at the last joint of the robotic arm. By evaluating the measured data, it can be concluded that the proposed vibration measurement methodology should be applicable.

The article is arranged in basic chapters and subchapters. The basic chapters include the Introduction, Design of Measurement Methodology, Measurements Performed, Evaluation of Measurements, Discussion and Conclusion.

## 2. Design of Measurement Methodology

This chapter focuses on the description of the proposed vibration measurement methodology and noise of the six-axis collaborative robotic arm UR10. At the beginning of the chapter, a robotic arm is described, including the introduction of the basic parameters not only of the description but also the parameters important for their own subsequent measurement. The measurement methodology describes the exact procedure that is important for the successful collection and storage of the measured data during vibration and noise measurements. Within the measurement methodology, the individual boundary sub-conditions that will be used in the actual measurement are defined [15,16,17].

### 2.1. Six -Axis Robotic Arm of UR10

The six-axis collaborative UR10 robotic arm is one of the many products of the Universal Robots company. This robotic arm, as the UR10 indicates, at its end can carry a load of up to 10 kg. It is important to realize that the weight of the end effector must be deducted from this weight. The workspace of the robot extends 1300 mm from the base joint. The advantage of this robotic arm is the possibility of hanging it on the side of the wall and the possibility of hanging it on the ceiling.

In Figure 1a is a sample of the robotic arm of UR10 and, in Figure 1b, the axes of the joints are marked (*q_x_*).

The limits of individual joints are important parameters for experimental solutions. These not only include the maximum angles of the individual joints, but also the maximum joint velocity. On Figure 2, the maximum angles of the rotation and velocities of the individual joints are given. Each joint has its name, given by the manufacturer. For this reason, the terminology of individual joints is also described [18,19,20,21,22].

### 2.2. Noise Measurement Methodology and Vibrations

As already mentioned, the contribution deals with the analysis of the movements of robotic devices focusing on the investigation of side effects, especially vibrations that generate robotic devices. The six-axis collaborative robotic arm UR10 is used for the solution. Current legislation has already solved the methodology to measure and evaluate noise and vibration in the working environment, but not how to measure and evaluate noise and vibration on specific devices. This subchapter is devoted to a description of the proposed methodology for vibration measurement, which is used in specific experimental measurements on the six-axis collaborative robotic arm UR10 in the laboratory conditions of the Faculty of Mechanical Engineering.

Experimental measurement at the velocity of individual joints of the robotic arm and the measurement of the overall noise of the robotic arm will be taken in several stages. The noise and vibration of each joint will be measured separately at different loads that will form the end effector at different velocities. A total of six positions of movements are determined for measuring the noise and vibration of individual joints. There is only one joint in each position. The range of joint movement is set in the range of 0 to 90°. At the end of the last joint is the rotary part, to which the end effectors are attached. The prerequisite is that the individual loads will have a rotary shape. Each measurement defined by the combination of the ocular conditions will be measured ten times in total, of which the arithmetic average will be reported in the evaluation of the experimental measurement.

#### 2.2.1. Joint Velocity

Each of the six joints (shown in Figure 2) has its maximum velocity. The joint velocities will be measured at 10, 20, 30, 40, 50, 60, 70, 80, 90 and 100% of the maximum velocity. Individual joints velocities at the percentage share of maximum velocities are shown in Table 1. The reason for choosing these percentage velocities is that we should be able to see the effect of velocity on the acceleration and noise of the robotic arm from the measured values. The percentage velocities of each joint will always be set on all joints of the robotic arm for a given measurement.

#### 2.2.2. Shoulder Load

In the subsection above, the maximum load capacity of the arm is given as 10 kg, including the weight of the end effector. This maximum load is, like velocity, divided into partial loads that are 1, 3, 5, 7 and 10 kg, including the attachment holder (effector). The measurement of the noise and vibration with individual stress will be measured again to move individual joints at the defined velocities above. Measurements will include an unprecedented arm. The reason for choosing load values is that we should be able to see the effect of the load on the acceleration and noise of the robotic arm from the measured values. We assume that the greater the load on the robotic arm, the greater the measured accelerations at each joint and the overall noise should be.

#### 2.2.3. Shoulder Position for Experimental Measurement

There have been six measurement positions to measure the load individual joints defined. Only one joint moves in each position. The movement of each joint is always from the initial position (color model) by 90° along a quarter-circle path to the position of the contour model. The trajectory for each joint is given by the axis distances between the joints performing the movement.
**Measurement Position****Joint movement****Description**1.
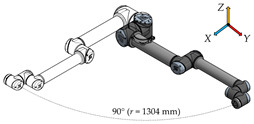
In the first measurement position, only the first joint of the robotic arm—the Base—moves. The axial distance between the Base and the outermost sixth joint of the arm—Wrist 3 is 1304 mm.2.
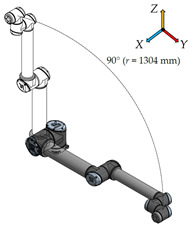
In this measurement position, only the second joint of the robotic arm—the Shoulder—moves. The longest distance is again 1304 mm and is determined by the axial distance between the second joint (Shoulder) and the sixth joint (Wrist 3) of the robotic arm.3.
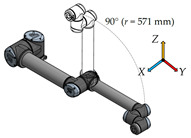
The movement between the third joint—Elboe and the sixth joint—Wrist 3 is the measurement for this position. The trajectory of movement between the joints is along the axial distance of 571 mm.4.
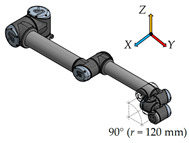
The shortest axial distance of 120 mm between the fourth joint—Wrist 1 and the sixth joint—Wrist 3 is for this measurement position.5.
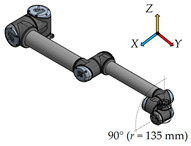
The second shortest distance of 135 mm between the fifth joint—Wrist 2, and the sixth joint—Wrist 3, is for this position6.
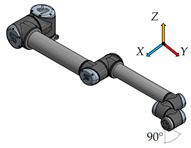
At the last sixth measurement position, the sixth joint of the robotic arm—Wrist 3—is measured.

It is important to measure the vibration not only when the joint is moving, but also on the other joints that are not performing any rotational motion at that moment. The reason for measuring vibration even on joints that do not perform motion is the transmission of vibration from the moving joint through the entire arm. Therefore, all six joints will be measured. The sensors (vibrometers, accelerometers) will be placed in the axis of the motor caps, as shown in Figure 3.

The noise from the movement of the robotic arm will be measured using two noise meters located in the *X* -axis at a distance of 50 cm from the most distant arm (Figure 4a) and at a height of 74 cm from the mounting foot in Figure 4b. It is approximately half of the maximum height of the arm in the *Y* -axis.

#### 2.2.4. Arrangement of Noise Measurement and Vibration

Vibration and noise measurements will be measured using two independent measurement devices. The vibrations will be measured by vibrometers and the Kistler measuring apparatus, and the noise will be measured by a Brüel&Kjær noise device and a Pulse measuring apparatus.

The vibration measuring string consists of six-axis 8316A2D10 accelerometers with ±10 g that will be connected to 19 “16 channel measuring Kidaq Rack from Kistler. The measuring chain for vibration is shown in Figure 5.

The noise of the robotic arm will be measured using two ½” microphones of type 4189-A-021 with a range of 6–20 kHz. These microphones will be connected to the 6-channel measuring units Pulse 3560-B-120 from the manufacturer Brüel&Kjær. The measurement and evaluation software will be in the notebook, to which the measuring unit will be connected. The measurement chain for the noise detector is shown in Figure 6.

The vibration and noise of the robotic arm will be measured 10 times every time, and percentage velocities of each measurement are measured. The mean value will be calculated from the measured values. The sub-velocities are always set on all joints of the robotic arm for a given measurement. A total of 500 noise and vibration measurements will be made—see Table 2.

## 3. Measurements Performed

In order to validate the measurement methodology, we used existing measurement equipment at our disposal. These are a three-axis accelerometer with TEDS, 10mV.m^−1^.s^−1^, type 4524-B, serial number 33154, and a six-channel PUlSE 3560-B-120 measuring unit from Brüel&Kjær. Using these devices, we measured the acceleration at the end of the sixth joint—Wrist 3—according to the first measurement position from the proposed vibration measurement methodology. The aim of the acceleration measurements is to verify whether the proposed measurement methodology can actually be used.

The measuring chain for measuring the acceleration is shown on Figure 7. It consists of a three-axis piezoelectric accelerometer with TEDS, 10mV.m^−1^.s^−1^, type 4524-B, No. 33154, which is connected to a six-channel measuring unit, PUlSE 3560-B-120. The PUlSE measuring unit is connected to a notebook, in which a measuring program for each measurement axis—*X*, *Y* and *Z*—was created in PUlSE software. The accelerometer was attached to the end effector (two-fingered gripper RG2) attached to the end of the sixth joint of the robotic arm (Wrist 3)—see Figure 8a. This was the robotic arm loaded with the mass of the end effector, whose mass is about 1 kg.

For the correct subsequent evaluation of the measured acceleration values in individual directions, it is important to know the direction of the axes in which the measurements were made. The direction of the axes can be seen in Figure 8b.

The measurement of acceleration according to the first position of the measurement methodology (the first joint of the robotic arm—Base) was taken for three velocities—20% (26.2°/s), 60% (78.6°/s), and 100% (131.0°/s). The reason for choosing these velocities was just to check whether the methodology could be implemented. In total, 10 measurements were taken for each velocity. The arithmetic average value was calculated from the measured data using MATLAB software.

## 4. Evaluation of Measurements and Discussion

Based on the data shown in Table 3, Table 4 and Table 5, the acceleration progress depends on the frequency in the individual accelerometer axes for the individual movement velocities of the first joint—the Base of the robotic arm. The first three dominant frequencies are listed for each measurement direction.

The graphs presented above show that the largest vibrations are in the *X*-axis. This fact is logical, since in this axis, the robotic arm performs a quarter-circle motion according to the already mentioned the first position of the proposed methodology, when the first joint (Base) of the six-axis collaborative arm performs the motion. Theoretically, the accelerations in the *Y* and *Z*-axis should be zero during the motion of the first joint of the robotic arm. However, we know from practice that a small amount of the acceleration is transferred to these axes.

The transmitted vibrations in the *Y*- and *Z*-axes are caused by the oscillation of the robotic arm construction due to the *X*-axis movement in the gearboxes of the individual joints of the robotic arm. Depending on the curves and results of the acceleration in the *Y*-axis and *Z*-axis, there are very low accelerations, but they accuracy affections and the repeatability of the robotic arm movements cannot be confirmed.

In Figure 9, Figure 10 and Figure 11 are acceleration progress depending on the frequency of *A_x_*, *A_y_* and *A_z_* in *X-*, *Y-* and *Z*-axes for velocity 20% (26.2°/s), 60% (78.6°/s) and 100% (131.0°/s).

From the graphs above where the acceleration vs. frequency waveforms are presented in each graph for each velocities, it is evident that the largest accelerations are indeed in the *X*-axis, where the robotic arm is moving.

In Figure 12, Figure 13 and Figure 14, a comparison of the acceleration waveforms *A_x_*, *A_y_* and *A_z_* in *X-*, *Y-* and *Z*-axis frequencies for velocities of 20% (26.2°/s), 60% (78.6°/s) and 100% (131.0°/s) is shown.

Table 6, Table 7 and Table 8 show the evaluated frequency values and the corresponding accelerations for each velocity (20, 60 and 100%) in the *X-*, *Y-* and *Z*-axes.

From the evaluated measured data, it can be seen that as the velocity of the robotic arm increases, the acceleration values also increase. In the *X*-axis, where the acceleration values are the highest compared to the *Y-* and *Z*-axes, the difference in acceleration between 20 and 60% of the velocity is 0.13996 m/s^2^. Between 60 and 100 % of velocity, the difference in the *X*-axis is 0.72105 m/s^2^. In the *Y*-axis, the difference in acceleration between 20 and 60 % of velocity is 0.01438 m/s^2^. Between 60 and 100 % of velocity, the difference in the *Y*-axis is 0.04949 m/s^2^. In the *Z*-axis, the difference in acceleration between 20 and 60% of velocity is 0.00134 m/s^2^. Between 60 and 100% of velocity, the difference in the *Z*-axis is 0.01381 m/s^2^.

The differences in acceleration values between 20, 60 and 100% of velocity show that this is not a linear dependence. The dependence cannot be determined yet, given that there are not enough measured data; it is only an initial measurement to verify the proposed methodology.

When creating the methodology of measurement of vibrations, we were looking for publications of authors who also focus on the issue. Hu et al. [23] in their contribution deal with optimal sensor placement method for vibration signal acquisition in the field of industrial robot health monitoring and fault diagnosis. For optimum sensor locations, the authors used the FEM (Finite Element Method), using ANSYS Workbench software. Using this method, they numerically detected the first six of their own oscillation shapes to them. At the same time, they found out which parts of the robotic arm were the most stressed. The aim of this solution was to optimize a total of sixteen positions where to place the sensors.

In their article, Bottin et al. discuss modeling and identification of an industrial robot with a selective modal approach [24]. In their article, they measure the vibration of the robot at ten test points. Test point number 1 was located on the flange of the end effector, and test point number 10 was located on the base of the robot. They created five test configurations for the measurement. They verified their experimental measurements using numerical simulations.

The dynamic and vibration characteristics of a newly designed long-range five-axis robotic arm for farming applications were discussed in the paper by Badkoobehhezaveh et al. [25]. In the article, the five-axis robotic arm was tested using Finite Element Method (FEM) and experimentally. The FEM was performed for two different manipulator configurations (for a fully extended manipulator and a half-extended manipulator). The FEM results showed that the first six natural frequencies of the manipulator for the two configurations considered were between 4.4 and 41.6 Hz. In the experiments, the acceleration data were acquired by the sensors and were subsequently processed using Fast Fourier Transform (FFT). The results from FEM and experiments were compared with each other. It showed that the comparison results between FEM and experiments had good agreement with less than 10% difference.

Based on a study of all the literature presented in this article, we have proposed a vibration and noise measurement methodology for the UR10 six-axis collaborative robotic arm. From the evaluation of the measured data presented in the previous section, it can be concluded that the proposed vibration measurement methodology should be applicable for further experimental measurements. The noise and vibration measurements according to the proposed measurement methodology on the six-axis collaborative robotic arm will be carried out during 2023. This is due to the delivery date of the measuring equipment from Kistler, which should be in early 2023. In the next phase of the experimental measurement, we will also look at the remaining movement velocities of the individual joints of the UR10 robotic arm, and it should already be possible to determine what the dependency is. We would like to publish all the evaluated measurements in a subsequent article, which should build on this article and refer to the proposed measurement methodology presented in this article.

## 5. Conclusions

The first part of the article is devoted to the design of a methodology for vibration and noise measurement on the six-axis robotic arm UR10. Prior to the actual design of the methodology, a search was carried out to see if a methodology for measuring vibration and noise on robotic arms already exists. We have found that there is no comprehensive up-to-date legislative methodology that describes exactly how to deal with the issue. The legislation so far only addresses the impact of vibration and noise on personnel moving in the environment where robotic arms are working. The vibrations generated by the movements of the robotic arms have a significant impact on the accuracy and repeatability of the tasks performed by the robotic arms. The proposed methodology would experimentally determine the amount of vibration generated in each of the six joints of the robotic arm during its movements. For the experimental measurements, a total of six positions in which the vibrations will be measured under different boundary conditions have been designed. The boundary conditions include different loads on the robotic arm and different movement velocities of each joint of the robotic arm. In parallel with the vibration measurements, the overall noise of the robotic arm will be measured in two mutually perpendicular axes. As already mentioned, this article is an introduction to the subject, where the problem is described for the time being. All measurements will be performed during 2023. The results will be discussed in the following article.

In the next part of the article, the initial measurement is described. In the initial measurement, the accelerations and their corresponding frequencies were measured for the first position of the proposed vibration measurement methodology. The measurements were performed while loading the end of the robotic arm with the weight of the gripper with an approximate weight of 1 kg. The velocities of the robotic arm motion were chosen as partial percentages of 100% of the velocity of the first joint of the arm. These were 20%, 60% and 100% of the joint velocity. The choice of these velocities was only for the purpose of verifying whether the proposed methodology can be applied. As a result, the acceleration versus frequency plots were plotted for the given velocity. From the measurements, it was found that vibrations are transmitted in directions other than the direction of motion. This fact may have a significant effect on the accuracy and repeatability of the movements when manipulating the robotic arm. The largest accelerations were measured in the *X*-axis, where the robotic arm performed the motion. In the *X*-axis, the acceleration difference between 20 and 60% of the velocity is 0.13996 m/s^2^. Between 60 and 100% velocity, the difference in the *X*-axis is 0.72105 m/s^2^. In the *Y*-axis, the difference in acceleration between 20 and 60% velocity is 0.01438 m/s^2^. Between 60 and 100% velocity, the difference in the *Y*-axis is 0.04949 m/s^2^. In the *Z*-axis, the difference in acceleration between 20 and 60% velocity is 0.00134 m/s^2^. Between 60 and 100% velocity, the difference in the Z-axis is 0.01381 m/s^2^. The differences in acceleration values between the 20, 60 and 100% velocities show that this is not a linear relationship. We will hopefully find out what kind of dependence it is in further measurements.

Initial experimental measurements verified that the proposed methodology should work. However, only after all measurements will be performed it will be verified whether this is the case or not.

## Figures and Tables

**Figure 1 sensors-23-01629-f001:**
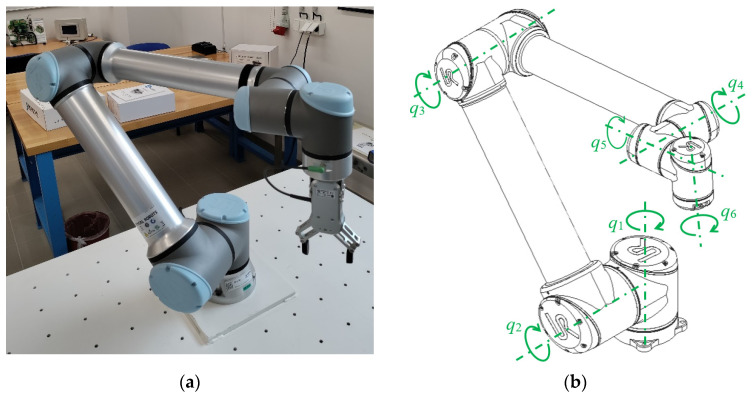
Six -axis collaborative robotic arm UR10: (**a**) photo of the robotic arm; (**b**) scheme of a robotic arm with marked joint axes.

**Figure 2 sensors-23-01629-f002:**
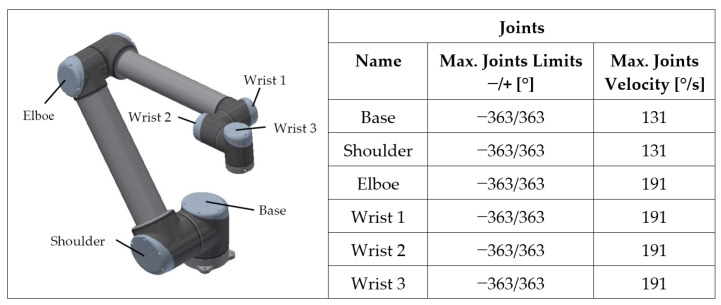
The names of individual joints, their maximum limits of rotation and velocity.

**Figure 3 sensors-23-01629-f003:**
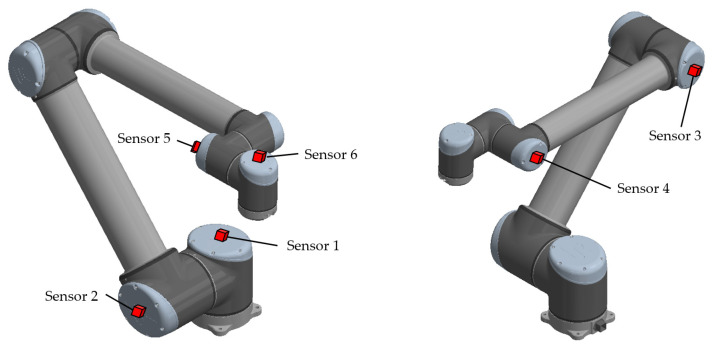
Location of sensors for vibration measurement.

**Figure 4 sensors-23-01629-f004:**
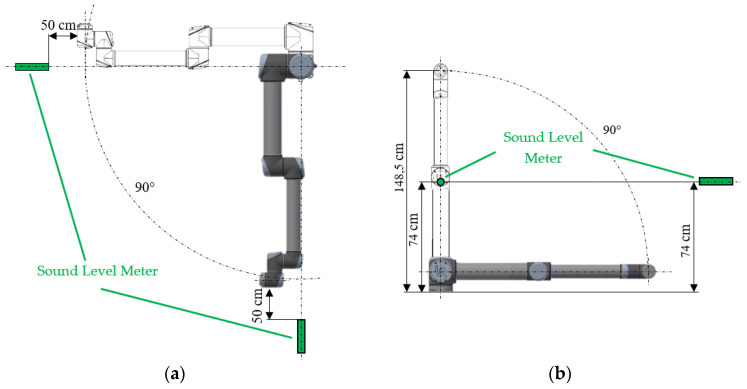
Location of noise measurement: (**a**) noise positions in *X*-axis; (**b**) the position of noise gauge in the *Y*-axis.

**Figure 5 sensors-23-01629-f005:**
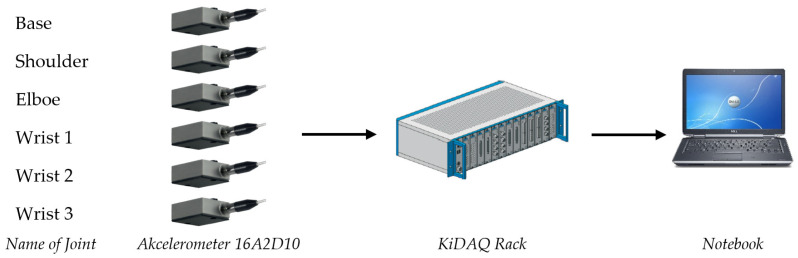
Measuring chain for vibration measurement.

**Figure 6 sensors-23-01629-f006:**
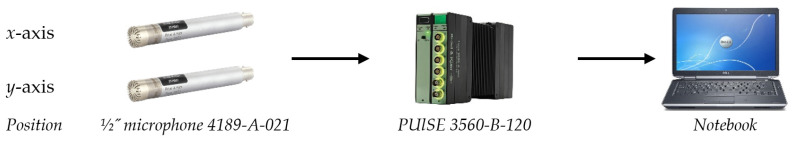
Measuring chain for noise measurement.

**Figure 7 sensors-23-01629-f007:**
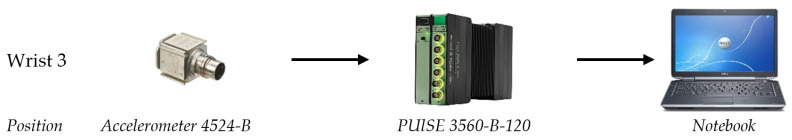
Measuring chain for measuring acceleration.

**Figure 8 sensors-23-01629-f008:**
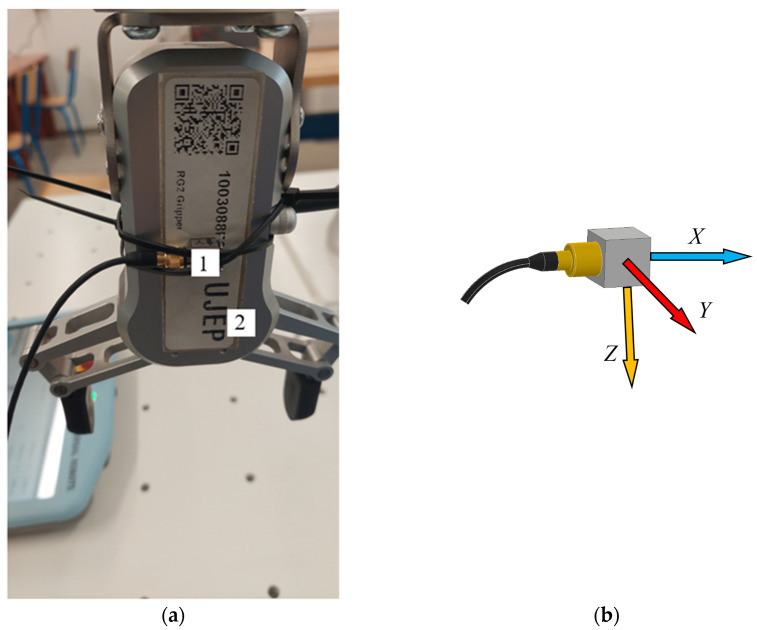
Accelerometer attachment—1 on a two-fingered gripper RG2—2 (**a**), directions of measurement axis on accelerometer (**b**).

**Figure 9 sensors-23-01629-f009:**
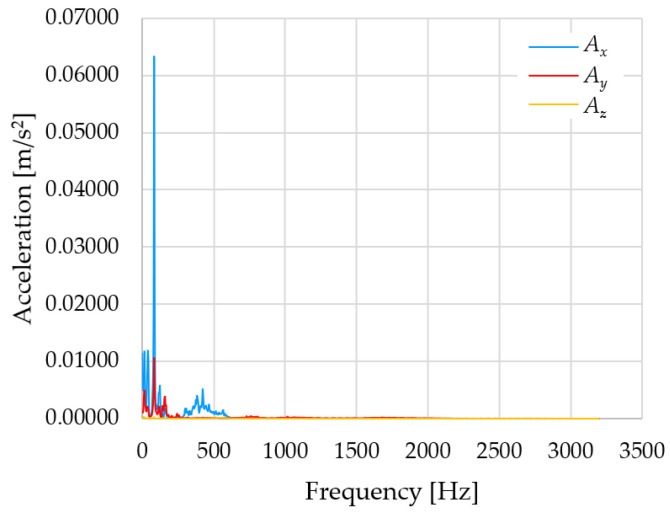
Acceleration progress depending on frequency in *X-*, *Y-* and *Z*-axes for velocity 20% (26.2°/s).

**Figure 10 sensors-23-01629-f010:**
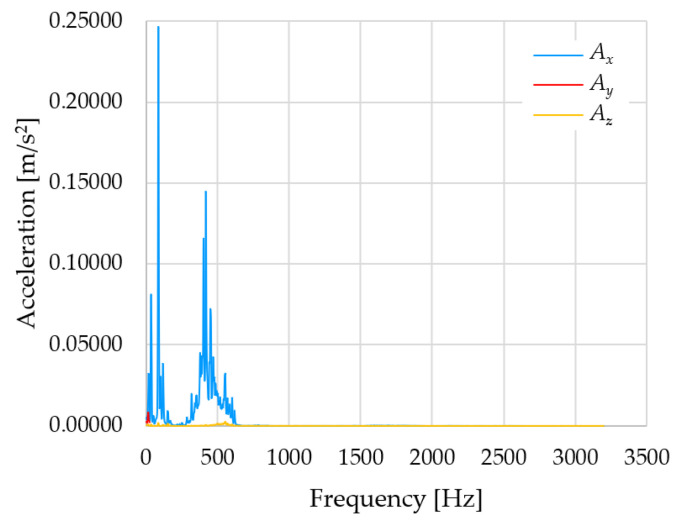
Acceleration progress depending on frequency in *X-*, *Y-* and *Z*-axes for velocity 60% (78.6°/s).

**Figure 11 sensors-23-01629-f011:**
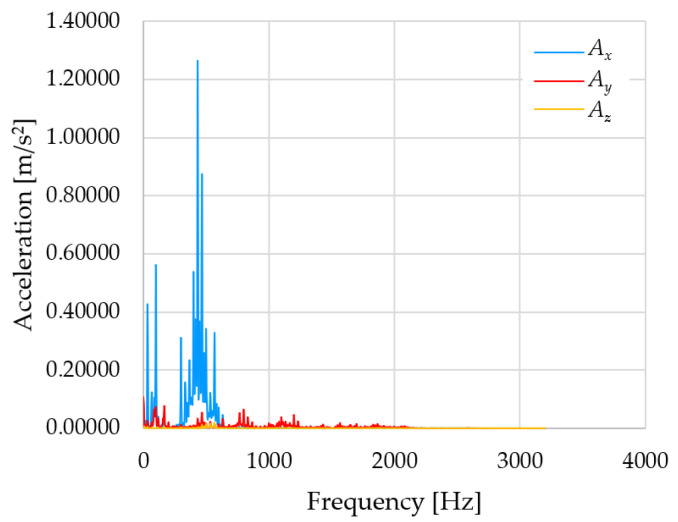
Acceleration progress depending on frequency in *X-*, *Y-* and *Z*-axes for velocity 100% (131.0°/s).

**Figure 12 sensors-23-01629-f012:**
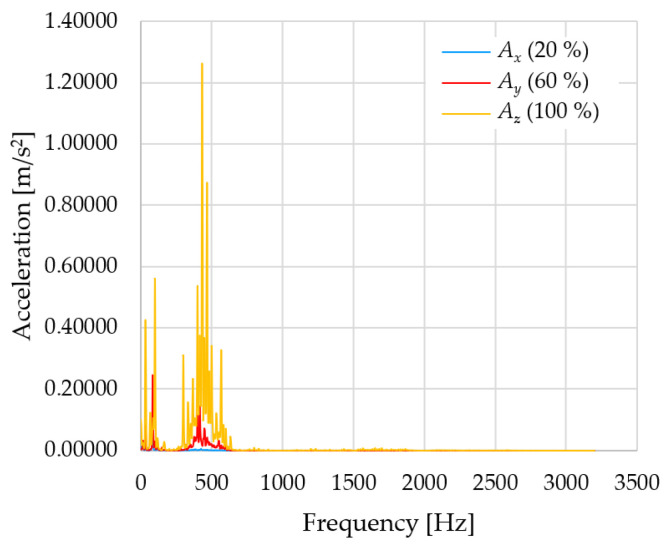
Acceleration waveforms depending on frequency in *X*-axis for velocity 20, 60 and 100%.

**Figure 13 sensors-23-01629-f013:**
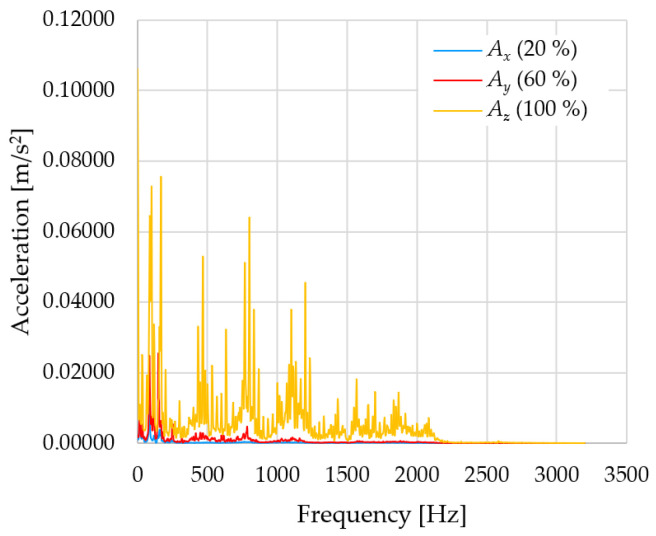
Acceleration waveforms depending on frequency in *Y*-axis for velocity 20, 60 and 100%.

**Figure 14 sensors-23-01629-f014:**
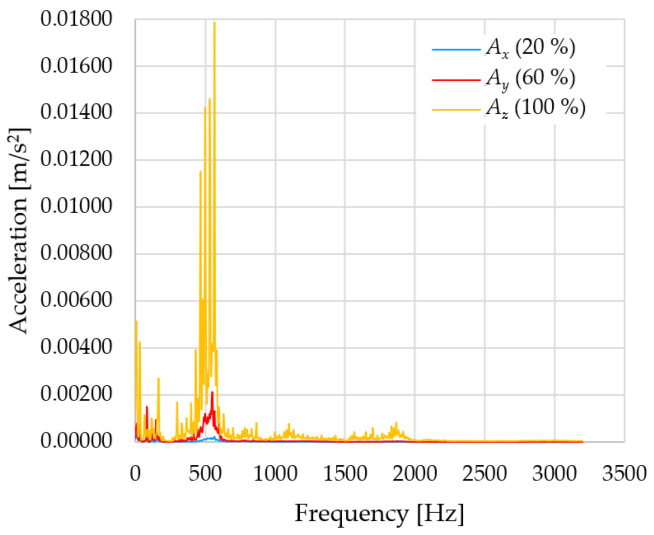
Acceleration waveforms depending on frequency in *Z*-axis for velocity 20, 60 and 100%.

**Table 1 sensors-23-01629-t001:** Percentage velocities from maximum velocity for individual joints.

Name of Joint	Joint Velocity [°/s]									
	10%	20%	30%	40%	50%	60%	70%	80%	90%	100%
Base	13.1	26.2	39.3	52.4	66.5	78.6	92.7	105.8	118.9	131.0
Shoulder	13.1	26.2	39.3	52.4	66.5	78.6	92.7	105.8	118.9	131.0
Elboe	19.1	38.2	57.3	76.4	96.5	115.6	134.7	153.8	172.9	191.0
Wrist 1	19.1	38.2	57.3	76.4	96.6	115.6	134.7	153.8	172.9	191.0
Wrist 2	19.1	38.2	57.3	76.4	96.6	115.6	134.7	153.8	172.9	191.0
Wrist 3	19.1	38.2	57.3	76.4	96.6	115.6	134.7	153.8	172.9	191.0

**Table 2 sensors-23-01629-t002:** Number of measuring the velocity and noise of the robotic arm.

Joint Velocity	Arm Load					
	1 kg	3 kg	5 kg	7 kg	10 kg	Total
**10%**	10	10	10	10	10	50
**20%**	10	10	10	10	10	50
**30%**	10	10	10	10	10	50
**40%**	10	10	10	10	10	50
**50%**	10	10	10	10	10	50
**60%**	10	10	10	10	10	50
**70%**	10	10	10	10	10	50
**80%**	10	10	10	10	10	50
**90%**	10	10	10	10	10	50
**100%**	10	10	10	10	10	50
**Total**	100	100	100	100	100	**500**

**Table 3 sensors-23-01629-t003:** The course of acceleration depending on the frequency for velocity 20 % (26.2°/s).

Accelerometer with Marked Axes	Graphs	The Dominant Frequency and the Corresponding Acceleration
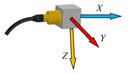	*X*-axis	
	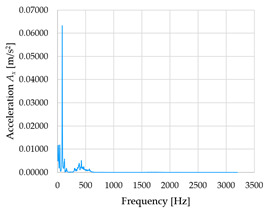	*F*_1_ = 16 Hz ≙ 0.01173 m/s^2^ *F*_2_ = 40 Hz ≙ 0.01187 m/s^2^ *F*_3_ = 84 Hz ≙= 0.06329 m/s^2^
	*Y*-axis	
	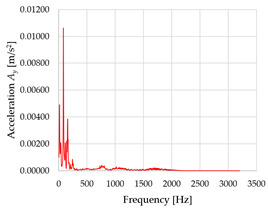	*F*_1_ = 16 Hz ≙ 0.00487 m/s^2^ *F*_2_ = 84 Hz ≙ 0.01064 m/s^2^ *F*_3_ = 160 Hz ≙= 0.00384 m/s^2^
	*Z*-axis	
	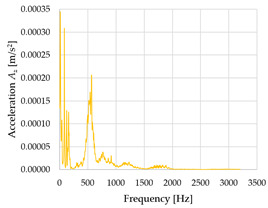	*F*_1_ = 12 Hz ≙ 0.00034 m/s^2^ *F*_2_ = 84 Hz ≙ 0.00031 m/s^2^ *F*_3_ = 564 Hz ≙= 0.00021 m/s^2^

**Table 4 sensors-23-01629-t004:** The course of acceleration depending on the frequency for velocity 60 % (78.6°/s).

Accelerometer with Marked Axes	Graphs	The Dominant Frequency and the Corresponding Acceleration
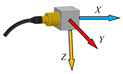	*X*-axis	
	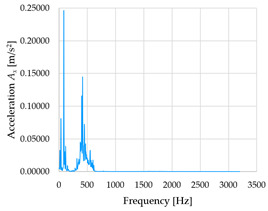	*F*_1_ = 84 Hz ≙ 0.24648 m/s^2^ *F*_2_ = 400 Hz ≙ 0.11576 m/s^2^ *F*_3_ = 416 Hz ≙ 0.14451 m/s^2^
	*Y*-axis	
	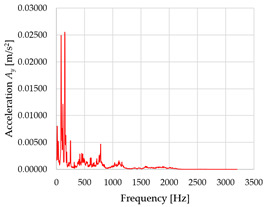	*F*_1_ = 84 Hz ≙ 0.02490 m/s^2^ *F*_2_ = 116 Hz ≙ 0.01214 m/s^2^ *F*_3_ = 148 Hz ≙ 0.02546 m/s^2^
	*Z*-axis	
	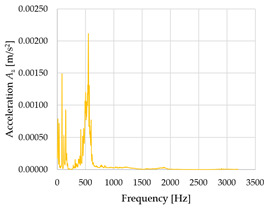	*F*_1_ = 84 Hz ≙ 0.00149 m/s^2^ *F*_2_ = 540 Hz ≙ 0.00131 m/s^2^ *F*_3_ = 552 Hz ≙ 0.00210 m/s^2^

**Table 5 sensors-23-01629-t005:** The course of acceleration depending on the frequency for velocity 100 % (131.0°/s).

Accelerometer with Marked Axes	Graphs	The Dominant Frequency and the Corresponding Acceleration
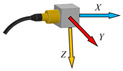	*X*-axis	
	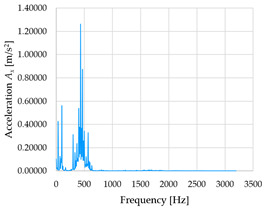	*F*_1_ = 100 Hz ≙ 0.56140 m/s^2^ *F*_2_ = 432 Hz ≙ 1.24956 m/s^2^ *F*_3_ = 468 Hz ≙ 0.85894 m/s^2^
	*Y*-axis	
	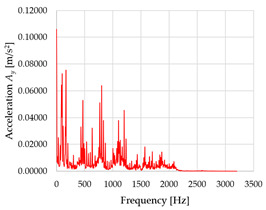	*F*_1_ = 88 Hz ≙ 0.06430 m/s^2^ *F*_2_ = 100 Hz≙= 0.07259 m/s^2^ *F*_3_ = 168 Hz ≙ 0.07408 m/s^2^
	*Z*-axis	
	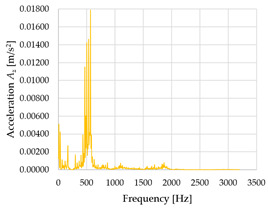	*F*_1_ = 500 Hz ≙ 0.01424 m/s^2^ *F*_2_ = 532Hz ≙ 0.01439 m/s^2^ *F*_3_ = 568 Hz ≙ 0.01770 m/s^2^

**Table 6 sensors-23-01629-t006:** Evaluated measured values of frequency and acceleration in *X*-axis for each velocity.

20% (26.2°/s)					60% (78.6°/s)				100% (131°/s)			
Tag.	Freq.	ø	Accel.	ø	Freq.	ø	Accel.	ø	Freq.	ø	Accel.	ø
	[Hz]	[Hz]	[m/s^2^]	[m/s^2^]	[Hz]	[Hz]	[m/s^2^]	[m/s^2^]	[Hz]	[Hz]	[m/s^2^]	[m/s^2^]
*F* _1_	16	46.66667	0.01173	0.02896	84	300.00000	0.24648	0.16891	100	333.33333	0.56140	0.88997
*F* _2_	40		0.01187		400		0.11576		432		1.24956	
*F* _3_	84		0.06329		416		0.14454		468		0.85894	

Legend: Freq.—Frequency, Accel.—Acceleration.

**Table 7 sensors-23-01629-t007:** Evaluated measured values of frequency and acceleration in *Y*-axis for each velocity.

20% (26.2°/s)					60% (78.6°/s)				100% (131°/s)				
Tag.	Freq.	ø	Accel.	ø	Freq.	ø	Accel.	ø	Freq.	ø	Accel.	ø	
	[Hz]	[Hz]	[m/s^2^]	[m/s^2^]	[Hz]	[Hz]	[m/s^2^]	[m/s^2^]	[Hz]	[Hz]	[m/s^2^]	[m/s^2^]	
*F* _1_	16	86.66667	0.00487	0.00645	84	116.00000	0.02490	0.02083	88	118.66667	0.06430	0.07032	
*F* _2_	84		0.01064		116		0.01214		100		0.07259		
*F* _3_	160		0.00384		148		0.02546		106		0.07408		

Legend: Freq.—Frequency, Accel.—Acceleration.

**Table 8 sensors-23-01629-t008:** Evaluated measured values of frequency and acceleration in *Z*-axis for each velocity.

20% (26.2°/s)					60 % (78.6°/s)				100% (131°/s)				
Tag.	Freq.	ø	Accel.	ø	Freq.	ø	Accel.	ø	Freq.	ø	Accel.	ø	
	[Hz]	[Hz]	[m/s^2^]	[m/s^2^]	[Hz]	[Hz]	[m/s^2^]	[m/s^2^]	[Hz]	[Hz]	[m/s^2^]	[m/s^2^]	
*F* _1_	12	220.00000	0.00034	0.00029	84	392.00000	0.00149	0.00163	500	533.33333	0.01424	0.01544	
*F* _2_	84		0.00031		540		0.00131		532		0.01439		
*F* _3_	564		0.00021		552		0.00210		568		0.01770		

Legend: Freq.—Frequency, Accel.—Acceleration.

## Data Availability

Not applicable.

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
