# Peer review of "Vibration Measurements on a Six-Axis Collaborative Robotic Arm—Part I"

_sensors, 2023, doi:10.3390/s23031629_

Round 1

Reviewer 1 Report

This is not a chapter. Somewhere you have written this chapter focuses?

Nomenclature should be added

In discussion, your result should be discussed 

English is very poor.

Why you are measuring the acceleration of the wrist. Why not each arm of the robot?

Why do you not consider all percentages of joint speed for finding the frequency and acceleration?

The conclusion is not clear. Try to elaborate.

Author Response

Dear Reviewer,
in the attached document, we are sending answers to your review.
Thank you.
Frantisek Klimenda et al.

Reviewer 2 Report

The main remarks:

1.The scientific purpose of the article is not clearly stated in the abstract, what does the presented work contribute to the development of research, what is the main contribution in the research field?

2. I encourage the author to reformulate the introduction section and describe clearly the contribution of the work related to other studies on this subject. Morever, it is necessary to inform about the Sections to be written in the paper, in the Introductin Section. 

3. The authors must reorganize the work more rigorously. For example, Section 4 is Measurement results Section and also Discussion Section.

4. Many  writing errors, for instances, Line "69" ..."However, vibrations are not as common as noise as noise",  Line "109"..''ter-minology"'

6. Figure 8 is not necessary. 

Author Response

(The authors gave the same response as above.)

Reviewer 3 Report

Dear authors,

you can find the review in the attached file.

Author Response

(The authors gave the same response as above.)

Round 2

Reviewer 2 Report

The authors responded concretely to the observations given by the reviewers.

Reviewer 3 Report

Dear authors,

after reading the revised manuscript, I can conclude, that you took all my comments into account.

The manuscript is well revised and improved.

I recomment to publish it in the recent form.